# Power generation from the interaction of a liquid droplet and a liquid membrane

Jinhui Nie [1,2], Ziming Wang[1,2], Zewei Ren[1,2], Shuyao Li[1,2], Xiangyu Chen [1,2] & Zhong Lin Wang [1,2,3]

Triboelectric nanogenerators are an energy harvesting technology that relies on the coupling effects of contact electrification and electrostatic induction between two solids or a liquid and a solid. Here, we present a triboelectric nanogenerator that can work based on the interaction between two pure liquids. A liquid–liquid triboelectric nanogenerator is achieved by passing a liquid droplet through a freely suspended liquid membrane. We investigate two kinds of liquid membranes: a grounded membrane and a pre-charged membrane. The falling of a droplet (about 40 μL) can generate a peak power of 137.4 nW by passing through a pre-charged membrane. Moreover, this membrane electrode can also remove and collect electrostatic charges from solid objects, indicating a permeable sensor or charge filter for electronic applications. The liquid–liquid triboelectric nanogenerator can harvest mechanical energy without changing the object motion and it can work for many targets, including raindrops, irrigation currents, microfluidics, and tiny particles.

---

[1] CAS Center for Excellence in Nanoscience, Beijing Institute of Nanoenergy and Nanosystems, Chinese Academy of Sciences, 100083 Beijing, China. [2] School of Nanoscience and Technology, University of Chinese Academy of Sciences, 100049 Beijing, China. [3] School of Materials Science and Engineering, Georgia Institute of Technology, Atlanta, GA 30332-0245, USA. Correspondence and requests for materials should be addressed to X.C. (email: chenxiangyu@binn.cas.cn) or to Z.L.W. (email: zhong.wang@mse.gatech.edu)

Since 2012, the triboelectric nanogenerator (TENG), which uses the coupling of contact electrification and electrostatic induction to generate electrical energy, has been gradually developing as a mainstream technology in the field of energy harvesting[1,2]. The operation of a TENG mainly relies on the interfacial electrostatic field, which outputs power due to the internal Maxwell displacement current induced by mechanical motion[3,4]. So far, TENGs can harvest energy from tribo-contacts at the interfaces of various substances[5], including the solid–solid interface[6], solid–liquid interface[7] and even the solid/liquid–air interface[8]. Targeting different mechanical energy sources, TENGs have been successfully applied in four major areas: micro/nano power sources[9,10], self-powered sensors[11,12], blue energy harvester[13], and high-voltage sources[14,15]. Among these areas, a TENG utilized at the solid–solid interface can help to achieve tactile sensing[16] and motion detection[17], and a TENG applied at the solid–liquid interface can produce energy from water waves[18], droplet motion[19,20] and so on. Although a large number of studies and applications have been performed based on TENGs and related systems, remarkably little progress has been proposed related to energy generation at the liquid–liquid (L–L) interface. During the contact motion between two liquids (droplets or streams), liquid objects are easily merged with each other, but their separation is difficult to realize with common experimental methods. Moreover, contact electrification on the L–L interface does not usually result in charge generation with opposite polarities on two separate interfaces, which is quite different from the solid–solid interface. Hence, the L–L interface remains undeveloped for researchers in the field of TENG and self-powered systems.

To achieve a smooth contact-separation motion between liquid objects, a highly durable and conductive liquid membrane is applied in our study. The penetrability, shape adaptability, and self-healing ability of the liquid membrane can effectively maintain a smooth separation of two liquids. The liquid membrane typically shows dynamic reconfigurability[21], stability, and elasticity[22], which can be used for antifouling transport[23], unusual particle separators[24], biomedical fluid handling[25] and so on. In recent years, such membranes have been shown to have extensive applications in biochemistry and medicine[26], environmental management, fossil fuel extraction, cosmetics and other industries. Nevertheless, the concept of utilizing liquid membrane as an electrification material is rarely mentioned, and it has never been developed as an active component for energy harvesting applications. A variety of properties for the liquid membrane can be achieved by adding different chemical components to the solution. The introduction of macromolecules in the solution can reduce the evaporation rate of the liquid membrane. In addition, thickeners are typically introduced to increase the surface viscosity of the solution and improve the elasticity of the liquid membrane within a certain range. By adjusting the surface tension, surface viscosity, volatilization rate of the solution, and the structure of the liquid membrane frame, the persistence of the liquid membrane can be extended for rather long periods of time, which allows this kind of material to have some practical applications.

Here, we show that an L–L TENG can be achieved by introducing a liquid membrane as a permeable electrode for contact with liquid droplets. Two kinds of liquid membranes have been developed for interacting with falling droplets. First, a grounded liquid membrane without charging is applied to collect the surface charge of falling droplets, which can harvest ambient electrostatic charges for energy generation. Second, a fluorinated ethylene propylene (FEP) film with tribo-charges is placed near the grounded liquid membrane, and a pre-charged membrane is achieved due to electrostatic induction. When water droplets pass through this pre-charged liquid membrane, the droplets carry away the charges from the membrane, and the charge redistribution induced on the liquid membrane can generate a current output. In this case, droplets with absolutely no surface charge can still generate energy from a pure L–L interface. Finally, the interaction between solid objects and the liquid membrane was also investigated, in which the liquid membrane can serve as a permeable sensor or charge filter for detecting and removing electrostatic charges on a solid surface. The proposed L-L TENG realizes the generation of displacement current on a pure L-L interface, which opens up a new direction for exploration for the study of TENG and self-powered systems.

## Results

**Structure of liquid–liquid triboelectric nanogenerator**. Figure 1a shows a dynamic contact-separation process between a water droplet and a liquid membrane. To maintain the surface tension of the liquid membrane, a three-dimensional (3D) printed annular frame is used to hold the membrane (inset of Fig. 1a). An electrical wire is integrated inside the groove of the annular frame, allowing the wire to directly contact with the liquid membrane for charge extraction. The penetrability and self-healing ability of liquid membrane facilitates the repeatable contact-separation motions of two liquid objects, and the conductive characteristic of the liquid membrane allows us to directly link the external circuit to the interaction interface. Hence, it is possible for us to achieve a different type of TENG specifically targeting the L–L interface. Relying on the penetrability of the liquid membrane, we can harvest mechanical energy without blocking or trapping the moving droplets, which is quite different from previous TENGs that were based on the electrification of a liquid–solid interface[19]. Hence, L–L TENG can work for many special targets, including raindrop, stream currents, and microfluidics. Figure 1b illustrates a schematic of a liquid membrane harvesting energy from raindrops without disturbing plant irrigation.

The lifetime of the liquid membrane is a critical factor for this device, and it has been systematically studied for several multicomponent systems. A liquid membrane can be obtained by mixing deionized water with various surfactants. Specifically, the lifetime of the liquid membrane when used with various surfactants, such as sodium dodecyl sulfate (SDS), sodium alcohol ether sulfate (AES), and the commercial soap of Walch, have been compared. The effects of different concentrations of sugar and polyvinyl alcohol (PVA) on the membrane lifetime were also compared (Supplementary Tables 1 and 2). In contrast, SDS is the better surfactant for the liquid membrane. Here, the sugar is used to reduce the evaporation rate (Supplementary Fig. 1a); as the sugar concentration increases (in the solution of SDS with a PVA concentration of 0.5 wt%), the rate of volatilization decreases. When the sugar concentration is ~5 wt%, the solution volatilization rate tends to be stable. Correspondingly, when the sugar concentration is ~5 wt%, the lifetime of the liquid membrane reaches a maximum (Fig. 1c). In addition, PVA is introduced into the solution as a thickener to improve the stability and lifetime, which can also be degraded successfully in the environment[27]. In the mixed solution with an SDS concentration of 0.3 wt% and a sugar concentration of 5 wt%, the viscosity of the solution increases as the concentration of PVA increases (Supplementary Table 3). As shown in Fig. 1d, the lifetime of a single membrane reaches a maximum value of more than 300 s with a PVA concentration of ~0.4 wt%. The lifetime is significantly reduced when the PVA concentration exceeds 0.4 wt%, and the surface viscosity is too large to maintain elasticity, which makes the liquid membrane unstable under the impact of the droplets. Finally, the

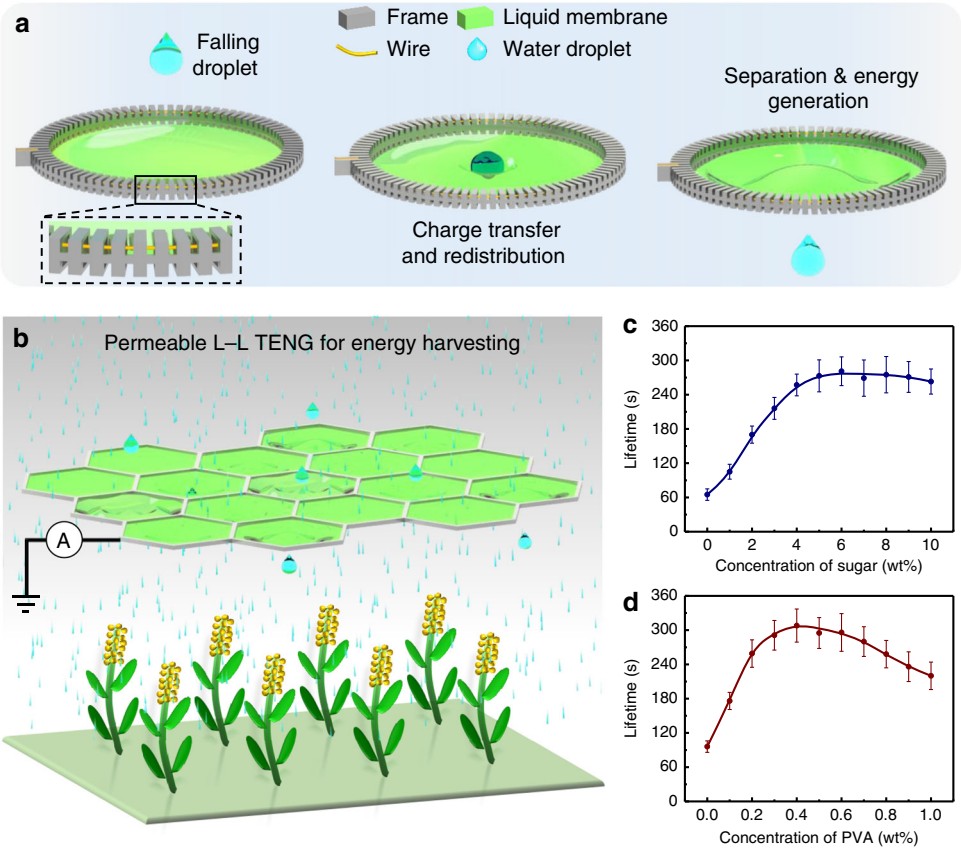

**Fig. 1** Concept of the liquid membrane for harvesting ambient electrostatic energy. **a** Schematics showing the working principle of the liquid membrane for collecting electrostatic energy from charged droplets. The inset shows structure design of the frame for holding liquid membrane. **b** Schematic diagram of L–L TENG collects energy from raindrops in an irrigation system, while the motion behavior of raindrop is unaffected. **c**, **d** The effect of different sugar concentrations (**c**) and different PVA concentrations (**d**) on the lifetime of the liquid membrane. Error bars represent standard deviation based on five replicate data. Test conditions: temperature 25 ± 0.5 °C, humidity 30 ± 2%, five drops of water per second through the liquid membrane. Source data are provided as a Source Data file

solution ratio of the membrane with the longest lifetime is 0.3 wt % SDS, 5 wt% sugar, and 0.4 wt% PVA in deionized water. The basic physical parameters of this solution are shown in Supplementary Table 4. Combined with the groove structure on the membrane frame to store the solution, the lifetime of the liquid membrane exceeds 5 min under the incessant impact of the water flow with a flow rate of $10\,mL\,s^{-1}$ (Supplementary Movie 1). Moreover, for practical applications, a reservoir structure (Supplementary Fig. 1b) is designed on the frame (Supplementary Movie 2). The main reason for the break of the liquid membrane is the evaporation of the solution on the membrane. Hence, the reservoir structure can help to replenish solution lost from the membrane during the repeated impact of droplets. With the help of the reservoir, a record high lifetime of over 5 h is realized under our lab environment by adding 1 mL of replenishing solution every 5 min.

**Falling droplets contacting a grounded liquid membrane.** During the falling process of a droplet, the triboelectrification between the droplet and air leads to tribo-charge accumulation on the droplet surface. Usually, positive charges are generated on the droplet, while the amount of induced charge is quite small. When the droplet contacts with the grounded liquid membrane, charges on the droplet are transferred to the liquid membrane. As seen in Fig. 2a, free electrons are attracted from the ground to neutralize the charges on the droplet. During the contact-separation process,

tiny substance exchange at the L–L interface can also occur, but this exchange does not influence the separation of the droplet. Due to the grounded membrane, almost all of the charges on the droplet should be neutralized. So far, based on our observations, the contact electrification on the L–L interface always leads to redistribution of the original charges on the two liquid objects due to the charge transfer effect and liquid fluidity. By utilizing this collection phenomenon for the liquid membrane, the first working mode of this L-L TENG can be established. To test the charge filtering performance, water droplets with different volumes fall from a height of 3 meters, and the liquid membrane is placed in the falling path to collect charges. Figure 2b, c shows the open-circuit voltage ($V_{OC}$) and short-circuit current ($I_{SC}$), respectively, with varying volumes of the falling droplets. With increasing droplet volume, increased surface area can lead to more adequate friction between water droplets and air. Hence, the output signal (both $V_{OC}$ and $I_{SC}$) increases with the increase of droplet volume. The electrical outputs due to the continuous flow of water droplets through the liquid membrane are shown in Fig. 2d, e, where water droplets with a volume of 40 μL fall from a height of 3 meters. In the experiment, the L-L TENG can produce a $V_{OC}$ of 33 mV, an average $I_{SC}$ of 0.85 nA, and a maximum charge transfer of 0.01 nC.

To study the velocity change for different sized droplets passing through the liquid membrane, the motion state of the droplets through the liquid membrane has been analyzed (Supplementary Fig. 2). Through force analysis, we list the

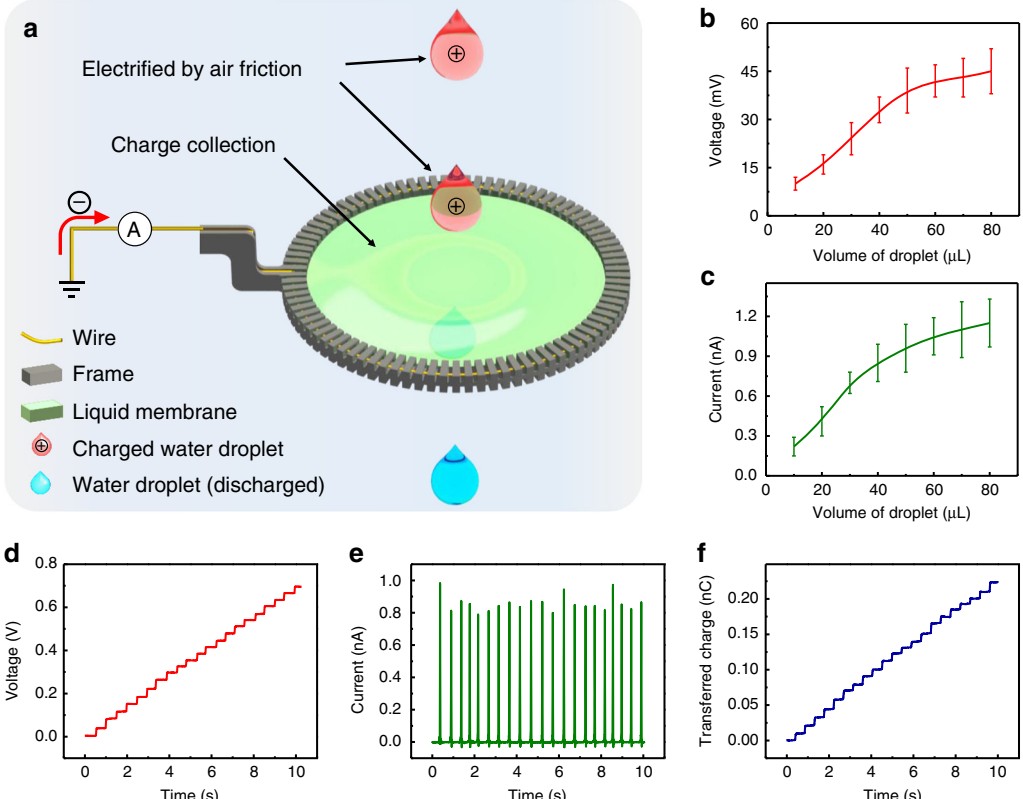

**Fig. 2** Falling droplets contacting with a grounded liquid membrane without pre-charging. **a** Schematic diagram of liquid membrane collects energy from falling droplets, while droplets carry positive tribo-charges due to the friction with air. **b, c** Open-circuit voltage ($V_{OC}$) (**b**) and short-circuit current ($I_{SC}$) (**c**) of droplets passing through the liquid membrane with different size. **d–f** $V_{OC}$ (**d**), $I_{SC}$ (**e**), and transferred charge (**f**) of droplets passing through the liquid membrane. Error bars represent standard deviation based on five replicate data. (All tests were performed with a drop height of 3 meters). Source data are provided as a Source Data file

following equation of motion according to Newton's second law:

$$G - F_s - F_a = ma \qquad (1)$$

where $G$ is the gravity of the droplet, $F_s$ is surface tension, $F_a$ is air resistance, $m$ is the mass of the droplet, and $a$ is the acceleration. $F_s$ and $F_a$ point in the opposite direction to $G$. Equation 1 can be expressed by the drop height ($h$) and drop time ($t$) of the droplet:

$$mg - \frac{4\pi\gamma}{R_d}R_\alpha^2 - \frac{1}{2}c\left(\frac{dh}{dt}\right)^2 \cdot \pi R_d^2 = m\frac{d^2h}{dt^2} \qquad (2)$$

where $g$ is the acceleration of gravity, $\gamma$ is the surface tension coefficient, $c$ is the air resistance coefficient, $R_\alpha$ is the radius of the circle where the droplet contacts the liquid film (which is a function of $h$), and $R_d$ is the radius of the droplet. The specific derivation of Eq. 2 and the expression of $R_\alpha$ with $h$ is explained in Supplementary Note 1. Based on the common raindrop speed[28], the velocity change for raindrops with varying diameter passing through the liquid membrane is calculated, as shown in Supplementary Table 5. When the diameter exceeds 0.4 mm, the raindrops can smoothly pass through the liquid film, and when the raindrop diameter is >1.2 mm, the liquid membrane reduces the speed of the raindrop by <1%. Based on statistical data, the common size of a raindrop is a few millimeters[29]; thus, this liquid membrane can allow the passage of almost all rain droplets, as detailed in the illustration shown in Supplementary Fig. 3a. Supplementary Fig. 3b shows a set of photographs that demonstrate the complete contact-separation process between a droplet and a liquid membrane. It is important to note that the calculation and analysis in Supplementary Note 1 is based on the

assumption that the surface tension ($F_s$) is homogeneously distributed on the surface of a droplet. If falling droplet is near the edge of the liquid membrane, the surface tensions on the two sides of the droplet are different, and a tangential force is induced on the droplet. We have recorded the motion behavior for a droplet passing through the edge region of a membrane, as seen in Supplementary Fig. 4. For larger droplets with a diameter of 3.6 mm, the motion trace for a droplet shows negligible change, and the unbalanced surface tensions lead to significant deformation of the droplet (see Supplementary Fig. 4a). For tiny droplets with a diameter of 0.6 mm, the unbalanced surface tensions can cause a significant lateral displacement for the droplet, and the induced lateral displacement point toward the center of the membrane (see Supplementary Fig. 4b). Hence, this kind of funneling effect of the liquid membrane automatically gathers droplets toward the center, which is good for maintaining the stability of the whole energy harvesting system. In comparison with previous TENGs based on the electrification of the solid–solid[30] or liquid–solid interface[31], the L–L interface brings negligible frictional force to the moving object, indicating less energy loss. Hence, most raindrops and common microfluidics can smoothly pass through the liquid membrane with their motion behavior almost unaffected.

**Falling droplets contacting a pre-charged liquid membrane.** For the previous application of the liquid membrane without charges, energy harvesting cannot occur if the falling object has absolutely no surface charge. Hence, in spite of its promising ability for charge collection, the membrane still needs to be

modified to fulfill the requirement of a nanogenerator. Charge accumulation on a liquid membrane can be easily induced by approaching an electrostatic field, and this characteristic was utilized to establish a modified L–L TENG for continuous energy generation. As shown in Fig. 3a, a pre-charged annular FEP film is placed around the liquid film by a frame, and the distance between them is ~1 cm to prevent discharge. Tribo-charges on the FEP film are induced by tribo-motion with a nylon film. This charged FEP film provides a high electric field for inducing

positive charges from the ground to the liquid membrane, and the small gap between the FEP film and liquid membrane can establish a capacitor for charge accumulation. The charging and discharging of this liquid membrane is realized by draining current from the ground, which is quite similar to the operation principle for a single electrode TENG[32]. The longitudinal section in Fig. 3b gives a step-by-step illustration for this liquid membrane working with the pre-charged mode. In the beginning, the negative tribo-charges on the FEP film and the positively charged

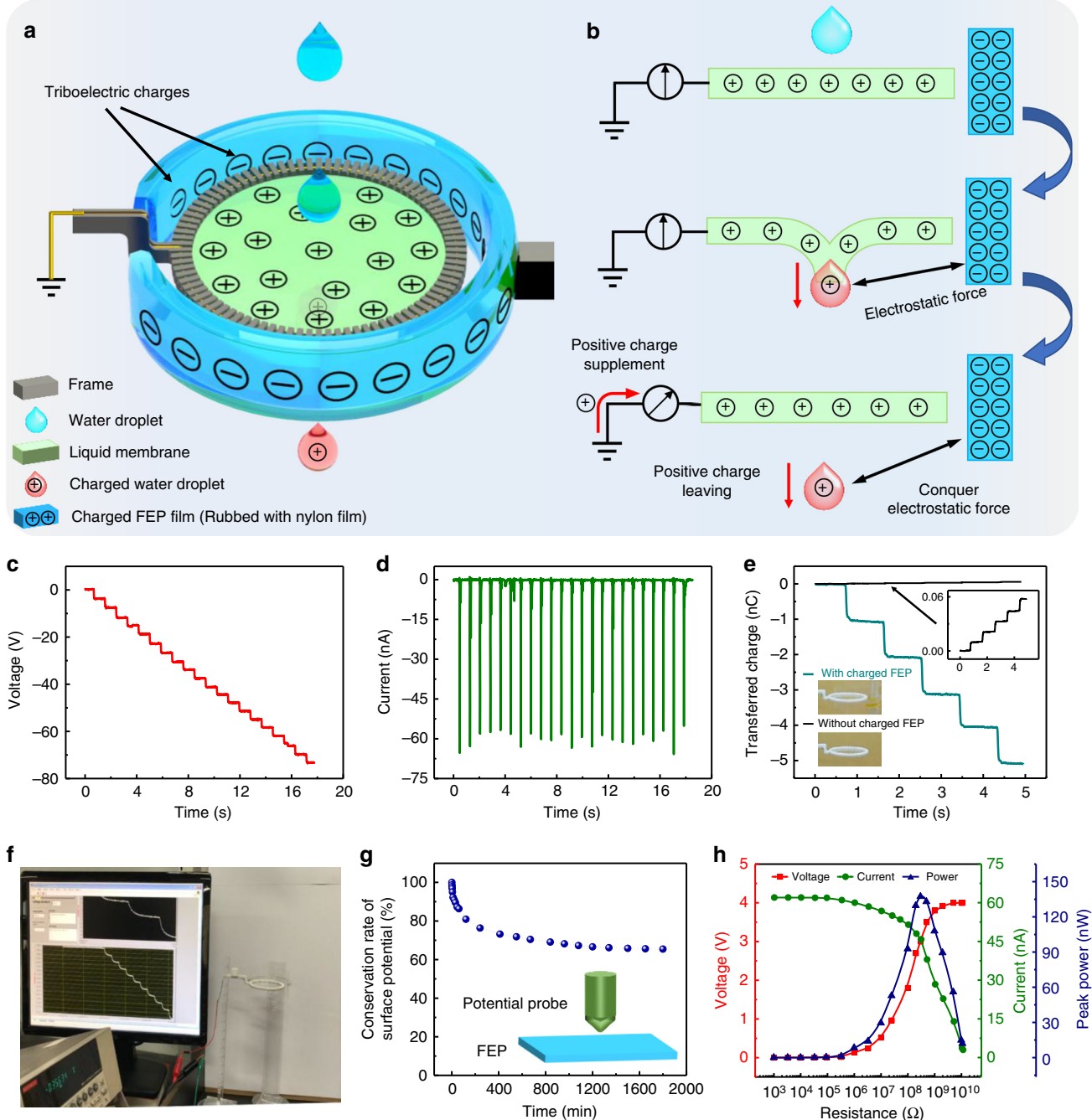

**Fig. 3** Falling droplets contacting with pre-charged liquid membrane. **a** Schematic diagram of energy generation by using a charged liquid membrane. The liquid membrane is charged by the electrostatic field on an FEP film. **b** Mechanism of energy generation of water droplets passing through the charged liquid membrane. **c, d** $V_{OC}$ (**c**) and $I_{SC}$ (**d**) of the droplets passing through the charged liquid membrane. **e** Transferred charge comparison of droplets passing liquid membrane with and without polarization. **f** Screenshot of the real-time output signals of the water droplets passing through the charged liquid membrane. **g** Stability of surface potential of the charged FEP film obtained by rubbing with nylon. **h** Dependence of the $I_{SC}$ and power on the resistance. Source data are provided as a Source Data file

liquid membrane establish a balanced system. When a water droplet (without surface charges) contacts with the liquid membrane, the positive charges are redistributed between the water droplet and the liquid membrane. Subsequently, the water droplet is separated from the liquid membrane, which takes away a certain amount of positive charges. Accordingly, the electrostatic balance is disturbed; thus, charges are required to be transferred from the ground to the membrane to compensate for the charges taken away by the falling droplets. Hence, a conduction current is induced from the ground to the liquid membrane, and energy generation is achieved. The entire system for this membrane-based nanogenerator can be simplified as a physical model with three capacitors (Supplementary Fig. 5a, b). The combined system of FEP film and liquid membrane is considered as a node because this system is far away from the ground. In the initial state, the FEP film and liquid membrane together form a balanced system, as shown in Supplementary Fig. 5a. At the open-circuit condition, when the droplet is separated from the liquid membrane, the effective charges on the droplet, balanced system, and ground are $q$, $-q$, and 0 (Supplementary Fig. 5b), respectively. It is important to note that we have assumed that a capacitance still exists between the droplet and the ground, even when the dropping height for the droplet is infinity. Thus, based on the basic characteristics of capacitance and charge conservation at each position, the maximum $V_{OC}$ can be obtained from the following equation:

$$V_{OC} \approx \frac{q}{2C_1} \qquad (3)$$

where $C_1$ is the capacitance between the balanced system and the ground. The factor 1/2 in Eq. 3 suggests only half of the effective charges on the liquid membrane can contribute to the $V_{OC}$ value, and the other half is still occupied by the mutual induction between droplet and liquid membrane. The detailed deduction process can be seen in the Supplementary Note 2. As the droplet moves away from the liquid membrane, it is subjected to an electrostatic force from the FEP film and liquid membrane. Supplementary Fig. 5c, d shows the force analysis of a charged water droplet passing through a charged liquid membrane, where the FEP is assumed to be a circular cylindrical film. The electrostatic force ($F_e$) on a water droplet is shown in Eq. 4:

$$F_e = \frac{qQ(h)}{4\pi\varepsilon_0} \frac{h}{(h^2 + R_f^2)^{\frac{3}{2}}} \qquad (4)$$

where $R_f$ is the radius of the FEP film, $\varepsilon_0$ is the permittivity of vacuum, $h$ is the falling distance of the droplet, and $Q(h)$ is the effective charge from the combined system of FEP film and liquid membrane, which is a function of $h$. The detailed derivation of Eq. 4 is shown in Supplementary Note 3. With the transfer of charge from the ground to the liquid membrane, the electrostatic balance between the FEP film and liquid membrane is gradually recovered ($Q(h)$ is decreasing), and the resultant electrostatic force on the droplet is gradually decreased to zero. Hence, we employ the effective charge $Q(h)$ to analyze the resultant electrostatic force applied on the falling droplet. From the droplet separation to the final position of the droplet ($h$ tends to infinity), the work ($W_e$) done by the electrostatic force on the water droplet is the electrical potential energy in the system, which can be expressed as $W_e = \int_0^\infty F_e \cdot dh$. By neglecting the energy loss caused by the friction force on the liquid membrane, the relationship between the gravitational potential energy ($E_g$), $W_e$ and kinetic energy ($E_k$) of the water droplets can be expressed by the

following equation:

$$E_g - W_e = \Delta E_k \qquad (5)$$

The electrical energy output between the liquid membrane and ground is converted from the mechanical energy $W_e$.

Figure 3c, d shows the $V_{OC}$ and $I_{SC}$ for droplets passing through the charged liquid membrane; $V_{OC}$ can reach 4 V, and the average $I_{SC}$ is 60 nA. In comparison with the results shown in Fig. 2d, e, significant improvements in electrical output can be achieved from the same liquid membrane with the help of an FEP tribo-film. To study the charge transfer at different positions of the membrane, we prepare a larger frame with a diameter of 12 cm and select three different positions on the membrane (see Supplementary Fig. 6a). The $V_{OC}$ value at different positions (A, B, and C) is measured respectively, as shown in Supplementary Fig. 6b. The lowest $V_{OC}$ value is obtained at the center position of the membrane, while the $V_{OC}$ gradually increases with the contacting position moving toward the edge. The FEP film with tribo-charges is placed near the edge of the frame; thus, the center part of the membrane has the lowest electric field. Based on this TENG, a jet of water passing through the liquid film can illuminate a light-emitting diode (LED) (Supplementary Movie 3). To clarify the detailed enhancement of the energy generation from the two working modes, the amount of transferred charges from the same liquid membrane system with or without the FEP film was measured, as shown in Fig. 3e. The average amount of transferred charges per drop reaches 1 nC from the liquid membrane in the pre-charged mode, which is 80 times larger than the output in the pure grounded mode. Figure 3f shows a photograph of the device structure and real-time display of the $V_{OC}$ measured for the pre-charged liquid membrane, in which a step increase in the voltage signal can be observed with the continuous falling of droplets. Moreover, the $V_{OC}$ of the device can be maintained for a rather long time without relaxation, as seen in Supplementary Movie 3. On the other hand, the operation of this liquid membrane with the charged mode is mainly determined by the surface potential of the FEP film. Hence, it is necessary to study the relaxation behavior of the triboelectric charges on the FEP film. The change in surface potential of an FEP film can be seen in Fig. 3g. Although the decay rate is fast at the beginning, the surface potential is finally saturated at a steady value, and the stable surface potential is approximately 60% of the initial value; this potential can be maintained for several days[33]. Figure 3h gives the dependence of the $V_{OC}$, $I_{SC}$ and peak power on external resistance. Since the internal resistance of the device is large, the maximum peak power is ~137.4 nW when the external resistance is ~300 MΩ. In our experiments, the falling droplets are produced by a dropper with a fixed surrounding environment, which can only partially simulate the raindrop condition. We have briefly tested energy generation from the real raindrops by using the grounded liquid membrane (see Supplementary Fig. 7). It is necessary to note that the detailed charge polarity on a real raindrop is decided by many factors, such as wind conditions, cloud conditions, thunder conditions, relative humidity and so on[34–36]. Hence, for the grounded membrane, if we want to harvest energy from a real raindrop, a rectifier circuit is necessary for normalizing the current signal. Meanwhile, the surface charges carried by a raindrop only produces a negligible influence on the output signal from the second working mode with a pre-charged liquid membrane.

By coupling the pre-charged membrane (Fig. 3a) and the pure grounded membrane (Fig. 2a) together, a combined energy generation system can be realized, as seen in Fig. 4a. The first pre-charged membrane generates energy based on the induction effect

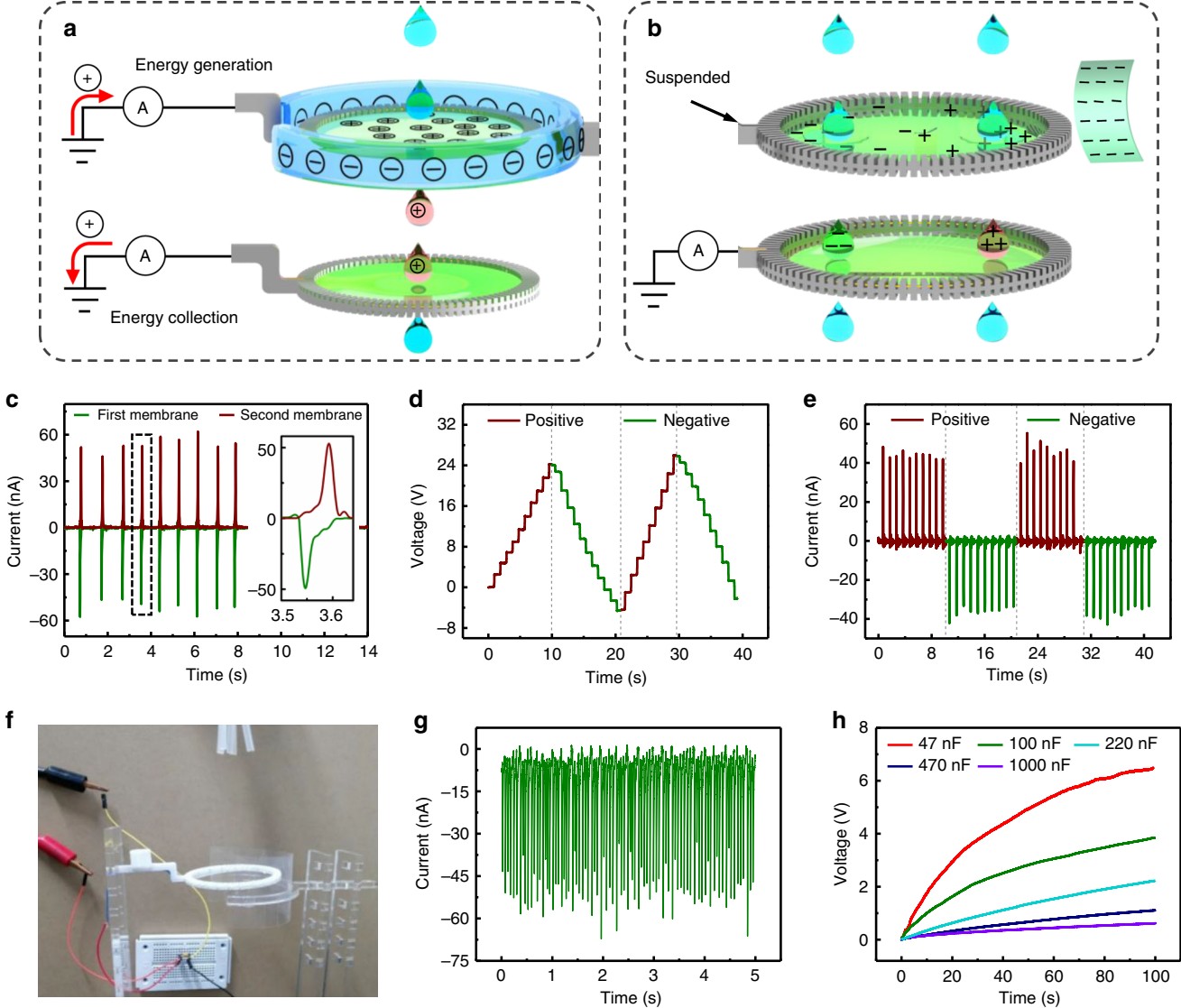

**Fig. 4** Power generations by passing droplets through multiple liquid membranes. **a** Schematic diagram of the multiple generations by combining a charged liquid membrane. **b** Schematic diagram of a suspended charged liquid membrane causing the passed droplets to be positively and negatively charged, respectively. **c** $I_{SC}$ of the droplets passing through the two liquid membranes in **a**. Insert is the output current of one droplet. **d, e** $V_{OC}$ (**d**) and $I_{SC}$ (**e**) of water droplets passing through different locations of the polarized liquid membrane. **f** Photograph of water droplets dripped simultaneously with five tubes for improving output power to charge capacitors. The drip rate of each tube is ~5 drops per second. **g** $I_{SC}$ of the multi-tube droplets passing through a polarized liquid membrane. **h** Charging voltage curves for different capacitors with the device in **f**. Source data are provided as a Source Data file

caused by FEP, and the second grounded membrane collects the charges on the falling droplet. Accordingly, one droplet can generate two pulse currents during the falling motion and no surface charge is wasted. Figure 4c shows the induced $I_{SC}$ from both the top and bottom membrane when the droplets pass through this combined system, and an enlarged figure for the current output within one cycle can be seen in the inset of Fig. 4c. The current signals through the two liquid membranes have opposite direction but similar amplitudes. By using this combined device, the output power from one droplet can be doubled. To demonstrate the scalability of the L-L TENG, we prepared a stacking TENG system with six liquid membranes to harvest energy from falling droplets. The pre-charged membrane and the grounded membranes were alternately arranged in the vertical direction, and the working principle of this stacking system is similar to that of the combined system in Fig. 4a. We have prepared a video to demonstrate the performance of this stacking system (see the third part of Supplementary Movie 3). Three

electrometers are applied to monitor the output signal from this stacking system, as seen in Supplementary Fig. 8a, b, and the output signal can be found in Supplementary Fig. 8c. A step increase in the amplitude of the voltage signal with the same frequency can be observed from both three electrometers, indicating that each liquid membrane can work normally. On the other hand, if an FEP film approaches a fully suspended liquid membrane, the liquid membrane will be polarized, with part of the membrane being positively charged and the other part being negatively charged, as shown in Fig. 4b. In this case, water droplets carry positive charges if they pass through the liquid membrane near the FEP film. Meanwhile, the droplets can also be negatively charged if they penetrate the other part of the membrane. The surface charges on the water droplet can be detected by the second membrane, and the corresponding $V_{OC}$ and $I_{SC}$ are shown in Fig. 4d, e. The amount of charge on the droplets passing through the different positions of the suspended liquid membrane is shown in Supplementary Fig. 9a. Hence, we

can control both the charge density and charge polarity on the droplets and such a liquid membrane can be applied for a microfluidic system or a transport system for tiny objects. Furthermore, to demonstrate the output capability, we prepare an experiment for charging different capacitors by using falling droplets. Five tubes are applied to provide droplets for a pre-charged liquid membrane (Fig. 4f). The total dripping rate of the droplet flow is ~20 drops per second and the induced $I_{SC}$ is shown in Fig. 4g. Correspondingly, the $V_{OC}$ can reach 240 V within 9 s (see Supplementary Fig. 9b). Figure 5h shows the voltage of different capacitors charged by this liquid membrane, where capacitors with a capacitance of 470 nF and 1000 nF can be charged to a voltage of 1.108 V and 0.616 V within 100 s, respectively. The charging efficiency of this system can be further improved by using a multimembrane system, as introduced in Fig. 4a.

**Solid object permeating through a grounded liquid membrane.** Not only the surface charges on the water droplets but also the surface charges on tiny solid objects can be collected by this liquid membrane, indicating several promising applications as a self-

powered sensor and a charge filter. When a solid substance passes through the liquid membrane, the surface of the substance is surrounded by this membrane, which causes the surface charges to be transferred through the liquid membrane. Figure 5a shows a set of photographs in which a circular acrylic block passes through a liquid membrane. Some common experimental consumables and tools can be easily charged after rubbing against rubber gloves, such as an acrylic block, polytetrafluoroethylene (PTFE) pellet, screwdriver, and even a pen. When these solid objects pass through the liquid membrane, the LED can be illuminated by the collected electrostatic charges (Fig. 5b–e and Supplementary Movie 4). The charge amount on the surface of the acrylic block, PTFE pellet (10 mm), screwdriver and pen is given in Fig. 5f. In addition, the charge amount of PTFE pellets with varying diameters is also shown in Fig. 5g, and the charge amount increases with increasing diameter. Other than falling objects, the liquid membrane can also be used to eliminate static electricity on the surface of tools (such as tweezers or screwdrivers), which can effectively prevent a microelectronic chip from being influenced by static electricity. On the other hand, the liquid membrane can be applied to detect the amount of charge

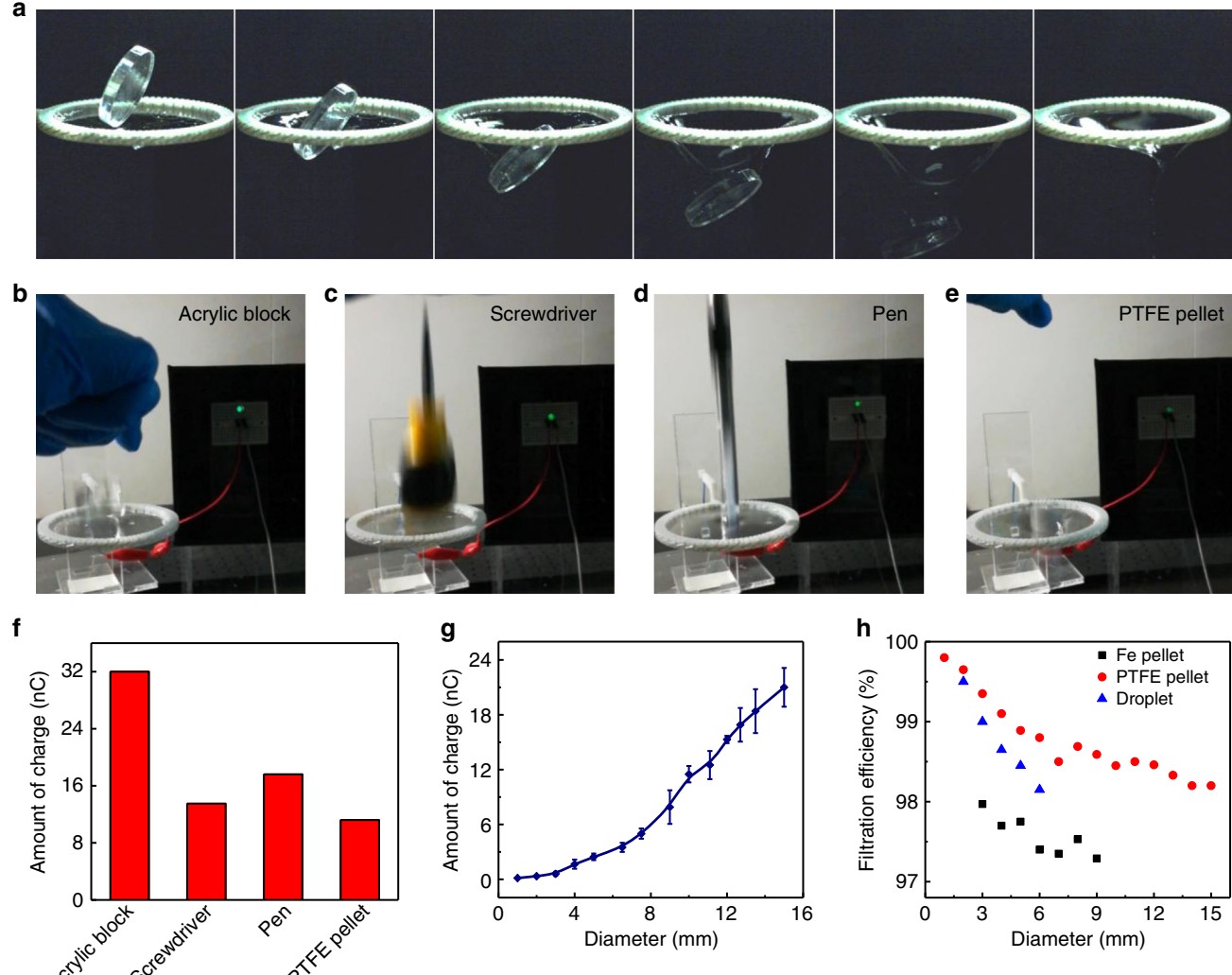

**Fig. 5** Liquid membrane collecting electrostatic charges from solid objects. **a** Images of an acrylic block passing through the liquid membrane. **b–e** Photographs of the liquid membrane filtering the electrostatic charges from the acrylic block, screwdriver, pen, and PTFE pellet, respectively. **f** Amount of charge of different objects filtered by the liquid membrane. **g** Amount of charge of PTFE pellets with different diameter filtered by the liquid membrane. **h** Charge filtration efficiency of the liquid membrane for Fe pellets, PTFE pellets and water droplets of different sizes. Error bars represent standard deviation based on three replicate data in panel **c**. Source data are provided as a Source Data file

carried by the object, and the surface charge density can be further calculated. Hence, the liquid membrane can serve as a self-powered sensor with high permeability for detecting ambient electrostatic charges.

As an electrostatic filter or an electrostatic sensor, the efficiency of charge filtration is an important parameter. To detect the filtration efficiency, a filtering system consisting of three liquid membranes was proposed for simultaneous detection (Insert of Supplementary Fig. 10a). Through multichannel detection, the $V_{OC}$ and $I_{SC}$ induced for each liquid membrane is recorded during the falling process of a droplet (Supplementary Fig. 10a, b). The current and voltage outputs for the first liquid membrane are more than 20 times larger than that for the second layer, and the third liquid membrane has almost no output. Moreover, the electrical signal induced during different objects (Fe pellet, PTFE pellet, and water droplet) through the three membranes was also recorded, and the filtration efficiencies of the first membrane for the different objects were calculated (Fig. 3h). For particles having a diameter of less than 15 mm, the charge filtration efficiency of the first liquid membrane exceeds 97%, and this efficiency can be very close to 100% after passing through a double-membrane system.

## Discussion

In this work, the interaction between falling droplets and a free-standing liquid membrane has been utilized to establish an L–L TENG, which can provide "continuous" energy generation from the L–L interface. By adjusting the chemical components in the solution, the lifetime of the liquid membrane can be larger than 5 min under the continuous impact of water flow (10 mL s$^{-1}$) or droplets dripping (5 drops per second). Moreover, with the help of a solution reservoir, the liquid membrane can keep working for hours without a break. Two kinds of liquid membranes (grounded and pre-charged) have been developed. The grounded membrane can remove the surface charges from liquid droplets, while energy generation can be realized during the charge collection process. For the pre-charged liquid membrane, an FEP film with negative tribo-charges can induce positive charges on liquid membrane. Then, the falling droplet takes away certain amount of charges from the pre-charged membrane and the displacement current is generated from ground to liquid membrane accordingly. The generated peak power of one droplet (with the volume of about 40 μL) reaches 137.4 nW. This output power can be further improved by applying multi-membrane system. On the other hand, this liquid membrane can also remove and collect electrostatic charges from solid objects. For tiny pellet having a diameter <15 mm, the charge filtration efficiency of the first liquid membrane exceeds 97%, and this efficiency can reach 100% by applying a double-membrane system. Hence, this liquid membrane can help to remove the electrostatic charge for various microelectronic apparatus, such as tweezers, screwdrivers, while it can also work as a permeable self-powered sensor for detecting surface charges. This work further expands the application scope of TENG by harvesting mechanical energy from pure L–L interface. Meanwhile, the utilization of liquid membrane as the permeable electrode for energy-related devices can also bring new ideas for the study of fluid physics and kinetics. The tirbo-motion on L–L interface encounters negligible friction force, indicating less energy loss in comparison with that on solid–solid or liquid–solid interface. Based on the permeability of liquid membrane, we can harvest mechanical energy without blocking or trapping the moving object, which can serve many targets, including raindrops, irrigation currents, microfluidics and tiny moving particles.

## Methods

**Lifetime of liquid membrane**. Three surfactants, AES (Aladdin), SDS (Aladdin), and Walch liquid soap (Valle), are selected to configure the solutions, and their concentrations are specified to be 0.3 wt%, 2.1 wt%, and 6 wt%, respectively. By adjusting the concentration of the thickener (PVA, Aladdin) and the sugar (Aladdin), the best matching concentration for the liquid membrane is obtained, and the concentrations of SDS, PVA, sugar are 0.3 wt%, 0.4 wt%, and 5 wt%, respectively. With a dripping rate of 5 drops per second (the drop height is 1 m), the lifetime of the liquid membrane exceeds 5 min. The specific adjustments and corresponding lifetimes can be seen in Supplementary Tables 1 and 2.

**Design of the three-dimensional frame**. To increase the lifetime, a frame for the liquid membrane is prepared by a 3D printing method. Polylactic acid (Stratasys) is selected as the printed material. A groove in the frame increases the storage of liquid, and connected grooves promote solution flow. The inner diameter, outer diameter, and thickness of the frame are 60 mm, 70 mm, and 5 mm, respectively. The depth of the groove is 2 mm, and the center angle corresponding to each groove is 2 degrees. Both the upper and lower sides of the frame have grooves. The grooves are connected by a hole with dimensions of 1 × 3 mm, and a bare electrical wire with a diameter of 0.5 mm is inserted into this hole. A reservoir with a volume of 1 mL is printed on the holder of the frame, which is connected to the grooves through an aisle on the holder.

**Measurement of the liquid membrane in grounded mode**. The electrical output performance is measured by a Keithley 6514 System Electrometer. The drop height of the droplets is 3 m. The water droplets are produced by a dropper. To test the charge filtration efficiency, a multichannel measurement system is used that includes a Keithley 6514 System Electrometer to simultaneously test the transfer charge of the three liquid membranes. The charge filtration efficiency ($\eta$) in Fig. 3h is calculated as $\eta = Q_1/Q_t$, where $Q_1$ is the transferred charge on the first liquid membrane, and $Q_t$ is the total charge on the droplet, which is obtained by adding the transferred charges for the three membranes.

**Measurement of the liquid membrane in pre-charged mode**. The FEP film used for polarizing the liquid membrane is 15 cm in length, 6 cm in width, and 150 μm in thickness. After rubbing against the nylon film, the FEP film is mounted on a frame and placed around the liquid membrane. The electrical output performance is measured by a Keithley 6514 System Electrometer. The surface potential of the FEP film is measured by a TREK 347 System Electrometer. The output performance for the combination of the filtering mode and the polarizing mode is measured by a double-channel measurement system with two Keithley 6514 System Electrometers.

## Data availability

The datasets generated during and/or analyzed during the current study are available from the corresponding author. The source data underlying Figs. 1c, d, 2b–f, 3c–e, g, h, 4c–e, g, h, and 5f–h and Supplementary Figs. 1a, 6b, 7, 8c, 9a, b and 10a, b are provided as a Source Data file.

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

## Acknowledgements

This work was supported by the National Key R&D Project from Minister of Science and Technology (2016YFA0202704), the National Natural Science Foundation of China (Grant Nos. 51775049, 51432005, 11674215, 5151101243, and 51561145021), the Beijing Municipal Science & Technology Commission (Z171100000317001), and Young Top-Notch Talents Program of Beijing Excellent Talents Funding (2017000021223ZK03).

## Author contributions

X.C. and Z.L.W. conceived the idea and supervised the experiment. X.C. and J.N. prepared the manuscript. J.N. and X.C. designed the structure of the device. J.N. performed the data measurements. Z.W., S.L. and Z.R. provided assistance with the experiments. All the authors discussed the results and commented on the manuscript.

## Additional information

**Competing interests:** The authors declare no competing interests.

