## [Peer Review File · Nature Communications]

Reviewers' Comments:

Reviewer #1:

Remarks to the Author:

The ms reported a novel liquid-liquid (L-L) TENG device achieved by passing liquid droplets through a freely suspended liquid membrane. Then authors studied the grounded membrane and pre-charged membrane with different passing through objects, and demonstrated charge collection. The research paper was well organized and presented in a proper style. I believe this ms is publishable and will have high impact in the TENG research field. However, it will be perfect that the authors could offer following information:

(1) Line 97, on page 3: "A liquid membrane can be obtained by mixing...."

Physical parameters of the liquid membranes are the foundation of this research. To better understand the membrane behaviors, it is desirable to present the liquid/membrane's parameters (not just composition ratio), such as density, viscosity, vapor pressure etc. for references.

(2) Line 128, on page 4: "Usually, positive charges are generated on the droplet...."

Is it possible to have solid evidence that the raindrops are positively charged (or negatively) prior to passing through the membrane ?

Reviewer #2:

Remarks to the Author:

The manuscript, entitled "Power generation from the interaction of liquid droplet and liquid membrane," was submitted for the publication of Nature Communications as an original article. The authors demonstrated triboelectrification using liquid-liquid contact for the first time using the interaction between falling droplets and a free-standing liquid membrane. In the light of social impact on the efficient energy harvesting, this work is worthwhile to be accepted for the publication especially because L-L TENG has never been demonstrated. However, there are a few points that should be addressed for the publication.

1. Figure 1b mislead the content. The liquid membrane can not be such large scale rectangular shape by comparing plants below the membrane. If the author want to schematically show the conceptual operation, it is better to show an array of small rings instead of large rectangular membrane.

2. In this kind of energy harvesting problem, "scalability" is always a critical issue. Thus, showing generated current through the stacking of rings more than 2 would be greatly strengthen the manuscript, while the performance of a pair of two membranes in Figure 4a and 4c was shown. Showing multiple pairs of rings of Figure 4a or 4c would be suitable results.

3. For investigating the mechanism, it is worth to demonstrate the falling drop passing through the membrane "not" at the center of membrane. Since falling droplet such as rain droplet may randomly fall through the membrane, passing different position rather than center may provide more clear physics related to the surface tension and electrical charge transfer.

Response to the referees' comments

We have revised the manuscript according to the referees' comments. Revised points and answers to the comments are summarized as follows:

Reviewer #1 (Remarks to the Author):

The ms reported a novel liquid-liquid (L-L) TENG device achieved by passing liquid droplets through a freely suspended liquid membrane. Then authors studied the grounded membrane and pre-charged membrane with different passing through objects, and demonstrated charge collection. The research paper was well organized and presented in a proper style. I believe this ms is publishable and will have high impact in the TENG research field.

Answer) Thank you very much for your kind consideration about our work. It is our great honor to have your affirmation.

However, it will be perfect that the authors could offer following information:

(1) Line 97, on page 3: "A liquid membrane can be obtained by mixing..."

Physical parameters of the liquid membranes are the foundation of this research. To better understand the membrane behaviors, it is desirable to present the liquid/membrane's parameters (not just composition ratio), such as density, viscosity, vapor pressure etc. for references.

Answer) Thank you very much for this suggestion. We measured the density, viscosity, vapor pressure, conductivity and surface tension of the optimized membrane solution (0.3 wt% SDS, 5 wt% sugar and 0.4 wt% PVA). Supplementary Table 4 is prepared to show the detailed parameters of this membrane solution. It is important to note that the lifetime of the membrane is the most important parameter for the energy harvesting application. Hence, the solution composition with longest membrane lifetime is the best choice for our experiment.

We explained this part in the manuscript, as can be seen on page 3, line 9 from bottom. "Finally, the solution ratio of the longest membrane lifetime is 0.3 wt% SDS, 5 wt% sugar, and 0.4 wt% PVA in deionized water. The basic physical parameters of this solution is shown in Supplement Table 4."

Supplementary Table 4. Physical parameters of the optimized membrane solution (0.3 wt% SDS, 5 wt% sugar and 0.4 wt% PVA). Test environment: Temperature 25 ± 0.3 °C, humidity 30 ± 2 %, atmospheric pressure 101.2 kPa.

Physical parameters	Values
Density (kg/m ³)	1018.5 ± 2.2
Viscosity (cP)	1.417

Vapor pressure (kPa)	0.363
Conductivity (μS)	559 ± 6
Surface Tension (mN/m)	33.57 ± 0.04

(2) Line 128, on page 4: “Usually, positive charges are generated on the droplet...”
 Is it possible to have solid evidence that the raindrops are positively charged (or negatively) prior to passing through the membrane?

Answer) Thank you very much for this inspiring comment. First of all, the falling droplets in our experiments are dripped out from a dropper. Based on our observation, the droplets produced in our lab environment are usually carrying positive charges and this may be different from the real raindrop. We have briefly tested the short-circuit current (I_{SC}) from the real raindrop by using the grounded liquid membrane. The results can be seen in Supplementary Figure 7. For the measurement of I_{SC} , the positive signals are more than the negative signals, and the current value is in the scale of 0.1~0.5 nA. The falling raindrop may carry some floating particles from air, which can change its surface charge polarity and raindrop stream may also be influenced by the electrostatic field of surrounding buildings. We have also tried the similar experiments at different locations with different rainfall amount but still cannot observe a unified charge polarity from raindrop.

On the other hand, there are a lot of previous works have studied the statistical data of the charges on raindrop^{1,2,3}. Generally speaking, the charge polarity on the raindrop can be decided by many factors (much more than what we can imagine), such as wind conditions, cloud conditions, thunder conditions, relative humidity and so on. For example, it has been found that more negatively charged droplets are observed in the developing stages of the cloud⁴. Meanwhile, positively charged raindrops predominated when the relative humidity was lower than 90%, whereas negatively charged particles prevailed at humidity over 90%⁵.

Hence, clarifying the charge polarity on real raindrop is beyond the study scope of this paper and we can offer some solutions for this problem from the viewpoint of energy harvesting. In this case, if we want to employ the grounded membrane to harvest energy from raindrop, a rectifier circuit is necessary for normalizing the current signal. Meanwhile, the surface charges carried by raindrop can only bring negligible influence to the output signal from the second working mode with pre-charged liquid membrane. As can be seen in Supplementary Figure 7 and Figure 3d, the charge amount carried by raindrop is quite small in comparison with the output signal from pre-charged liquid membrane. We further explained this part in the manuscript.

We explained this part in the manuscript, as can be seen on page 7, line 20. “In our experiments, the falling droplets are dripped out by a dropper with a fixed surrounding environment, which can only partially simulate the raindrop condition. We have briefly tested the energy generation from the real raindrop by using the grounded liquid membrane. The results can be seen in Supplementary Figure 7. It is necessary to note that the detailed charge polarity on the real raindrop is decided by many factors, such as wind conditions, cloud conditions, thunder conditions, relative humidity and so on^{34,35,36}. Hence, for the grounded membrane, if

we want to harvest energy from real raindrop, a rectifier circuit is necessary for normalizing the current signal. Meanwhile, the surface charges carried by raindrop can only bring negligible influence to the output signal from the second working mode with pre-charged liquid membrane.”

Supplementary Figure 7. I_{SC} of the real raindrops passing through the grounded liquid membrane.

Reviewer #2 (Remarks to the Author):

The manuscript, entitled “Power generation from the interaction of liquid droplet and liquid membrane,” was submitted for the publication of Nature Communications as an original article. The authors demonstrated triboelectrification using liquid-liquid contact for the first time using the interaction between falling droplets and a free-standing liquid membrane. In the light of social impact on the efficient energy harvesting, this work is worthwhile to be accepted for the publication especially because L-L TENG has never been demonstrated.

Answer) Thank you very much for your positive comments about our work. We are very grateful that you can accept the novelty of this work.

However, there are a few points that should be addressed for the publication.

1. Figure 1b mislead the content. The liquid membrane can not be such large scale rectangular shape by comparing plants below the membrane. If the author want to schematically show the conceptual operation, it is better to show an array of small rings instead of large rectangular membrane.

Answer) Thank you very much for this careful suggestion. We are deeply sorry for this misleading information. The Figure 1b has been revised.

Figure 1. Concept of the liquid membrane for harvesting ambient electrostatic energy.

2. In this kind of energy harvesting problem, “scalability” is always a critical issue. Thus, showing generated current through the stacking of rings more than 2 would be greatly strengthen the manuscript, while the performance of a pair of two membranes in Figure 4a and 4c was shown. Showing multiple pairs of rings of Figure 4a or 4c would be suitable results.

Answer) Thank you very much for this suggestion. In order to demonstrate the scalability of the L-L TENG, we prepared a stacking TENG system with six liquid membranes to harvest energy from falling droplets. The pre-charged membrane and the grounded membrane alternately arranged in vertical direction, while the working principle of this stacking system is similar as the combined system in Figure 4a. We have prepared a video materials to demonstrate the performance of this stacking system (see the third part of Supplementary Movie 3). Three electrometers are applied to monitor the output signal from this stacking system. The schematic diagram is shown in Supplementary Figure 8a and the corresponding photograph of the real system can be found in Supplementary Figure 8b. It is necessary to note that the droplet can carry a certain amount of charges after passing through the first pre-charged membrane. If these residual charges are not fully removed by the grounded membrane, they can disturb the energy generation happened on the other two pre-charged membranes and the electrometers may observe disordered signals. Hence, in our demonstration, the grounded membranes (the second, the fourth and the sixth) in this stacking system are working with short-circuit mode (connected to the grounding end of the electrometers). Supplementary Figure 8c shows the output signal of this staking system. The step increase of the amplitude of the voltage signal with the similar frequency can be observed from both three electrometers, suggesting that each liquid membrane in this system can work normally. Meanwhile, the impact process of droplets can also take away some solution from the liquid membrane, which leads to the increase of droplet volume. Hence, we can observe some sudden increase of voltage amplitude during the measurements, as shown in Supplementary Figure 8c.

We explained this part in the manuscript, as can be seen on page 7, line 5 from bottom. “In order to demonstrate the scalability of the L-L TENG, we prepared a stacking TENG system with six liquid membranes to harvest energy from falling droplets. The pre-charged membrane and the grounded membrane alternately arranged in vertical direction, while the working principle of this stacking system is similar as the combined system in Figure 4a. We have prepared a video materials to demonstrate the performance of this stacking system (see the third part of Supplementary Movie 3). Three electrometers are applied to monitor the output signal from this stacking system, as can be seen in Supplementary Figure 8a,b, and the output signal can be found in Supplementary Figure 8c . The step increase of the amplitude of the voltage signal with the same frequency can be observed from both three electrometers, indicating that each liquid membrane in this system can work normally.”

Supplementary Figure 8. A staking TENG system with six liquid membranes to harvest energy from falling droplets. a Schematic diagram of the staking TENG system with three electrometers monitoring the output signal. **b** Photograph of the test system. Insert is the close-up of the multiple pairs of membranes. **c** Output voltage of this staking TENG system. The impact process of droplets can also take away some solution from the liquid membrane, which leads to the increase of droplet volume. Hence, we can observe some sudden increase of voltage amplitude during the measurements, as shown in Supplementary Figure 8c.

3. For investigating the mechanism, it is worth to demonstrate the falling drop passing through the membrane “not” at the center of membrane. Since falling droplet such as rain droplet may randomly fall through the membrane, passing different position rather than center may provide more clear physics related to the surface tension and electrical charge transfer.

Answer) Thank you very much for this inspiring suggestion. It is important to note that the calculation and analysis in Supplementary Note 1 is based on the assumption that the surface tension (F_s) is homogeneously distributed on the surface of droplet. The falling droplet can induce a collapsing region on the liquid membrane, as shown in Supplementary Figure 3a. If the collapsing region is far away from the edge of the liquid membrane, the droplet will keep falling in the vertical direction after passing through the membrane and the calculation in Supplementary Note 1 is valid for these cases. However, if the edge of the liquid membrane overlaps the collapsing region induced by the falling droplet, the analysis based on Supplementary Figure 2 and Supplementary Note 1 need to be modified. The surface tensions on the two side of the droplet are different in this case, and the tangential force is induced on the droplet due to the unbalanced surface tensions. We have studied the motion behavior of droplet passing through the edge region of membrane by using high speed camera, as can be seen in Supplementary Figure 4. For the larger droplets with a diameter of 3.6 mm, the motion trace of the droplet show negligible change, while the unbalanced surface tensions

lead to the significant deformation of the droplet, as can be seen in Supplementary Figure 4a. For the tiny droplets with a diameter of 0.6 mm, the unbalanced surface tensions can cause a significant lateral displacement for the droplet and the induced lateral displacement is pointing at the center of the membrane (see Supplementary Figure 4b). Hence, this kind of “funneling” effect of liquid membrane automatically gathers droplets toward the center, which is good for maintaining the stability of the energy harvesting system.

As for the charge transfer process, we have studied the induced energy output of droplets passing through different region of the membrane, as shown in Supplementary Figure 6. In order to amplify the position difference, we prepared a larger frame with a diameter of 12 cm and selected three different positions on the membrane (see Supplementary Figure 6a). The V_{OC} value at different positions (A, B and C) is measured respectively, as shown in Supplementary Figure 6b. The lowest V_{OC} value is obtained at the center position of membrane, while the V_{OC} gradually increases with the contacting position moving towards the edge. The maximum voltage difference between the center position and the edge position is around 2 V. This effect is due the distribution of the electrostatic field. The FEP film with tribo-charges is placed near the edge of the frame and thus, the center part of the membrane has the lowest electric field.

We explained the surface tensions in the manuscript, as can be seen on page 5, line 4. “It is important to note that the calculation and analysis in Supplementary Note 1 is based on the assumption that the surface tension (F_s) is homogeneously distributed on the surface of droplet. The falling droplet can induce a collapsing region on the liquid membrane, as shown in Supplementary Figure 3a. If the edge of the liquid membrane overlaps the collapsing region induced by the falling droplet, the surface tensions on the two side of the droplet are different and the tangential force is induced on the droplet. We have recorded the motion behavior of droplet passing through the edge region of membrane, as can be seen in Supplementary Figure 4. For the larger droplets with a diameter of 3.6 mm, the motion trace of the droplet show negligible change, while the unbalanced surface tensions lead to the significant deformation of the droplet (see Supplementary Figure 4a). For the tiny droplets with a diameter of 0.6 mm, the unbalanced surface tensions can cause a significant lateral displacement for the droplet and the induced lateral displacement is pointing at the center of the membrane (see Supplementary Figure 4b). Hence, this kind of funneling effect of liquid membrane automatically gathers droplets toward the center, which is good for maintaining the stability of the whole energy harvesting system.”

Supplementary Figure 4. Motion behavior of droplets passing through the edge region of membrane with the size of 3.6 mm (a) and 0.6 mm (b).

We explained the charge transfer part in the manuscript, as can be seen on page 6, line 4 from the bottom. “In order to study the charge transfer at different position of membrane, we prepare a larger frame with a diameter of 12 cm and select three different positions on the membrane (see Supplementary Figure 6a). The V_{OC} value at different positions (A, B and C) is measured respectively, as shown in Supplementary Figure 6b. The lowest V_{OC} value is obtained at the center position of membrane, while the V_{OC} gradually increases with the contacting position moving towards the edge. This effect is due the distribution of the electrostatic field. The FEP film with tribo-charges is placed near the edge of the frame and thus, the center part of the membrane has the lowest electric field.”

Supplementary Figure 6. A larger frame is prepared for studying the different output at different falling position of the membrane. a A larger membrane for checking the charge transfer and three different positions on the liquid membrane for droplets to pass through. **b** V_{OC} of droplets passing through these positions.

References

1. Chauzy, S. & Despiau, S. Rainfall Rate and Electric Charge and Size of Raindrops of Six Spring Showers. *J. Atmos. Sci.* **37**, 1619-1627 (1980).
2. Takahashi, T. Measurement of Electric Charge of Cloud Droplets, Drizzle, and Raindrops. *Rev. Geophys. Space Phys.* **11**, 903-924 (1973).
3. Selvam, A. M., Manohar, G. K., Khemani, L. T. & Murty, B. V. R. Characteristics of Raindrop Charge and Associated Electric Field in Different Types of Rain. *J. Atmos. Sci.* **34**, 1791-1796 (1977).
4. Sergieva, A. P. Electrical Charges of Cloud Droplets. *Izv. Akad. Nauk SSSR, Ser. Geofiz.* **7**, 721-726 (1959).
5. Takahashi, T. Electric Charge of Small Particles. *J. Atmos. Sci.* **29**, 921-928 (1972).

Manuscript number: NCOMMS-19-05562

Title: " **Power generation from the interaction of liquid droplet and liquid membrane** "

Authors: Jinhui Nie, Ziming Wang, Zewei Ren, Shuyao Li, Xiangyu Chen, Zhong Lin Wang

Reviewers' comments:

Reviewer #1 (Remarks to the Author):

The authors well addressed my concerns in the revised manuscript. good work !

Reviewer #2 (Remarks to the Author):

The revised version clearly reflected the comments from my previous review and this manuscript is now ready to be published.

Reviewer #1 (Remarks to the Author):

The authors well addressed my concerns in the revised manuscript. good work !

Response:

We are grateful to the reviewer for taking the time to evaluate our work and support from the reviewer.

Reviewer #2 (Remarks to the Author):

The revised version clearly reflected the comments from my previous review and this manuscript is now ready to be published.

Response:

We thank the reviewer for taking the time to evaluate our work and support from the reviewer.